# Understanding the Eluder Dimension

**Gene Li**
Toyota Technological
Institute at Chicago
`gene@ttic.edu`

**Pritish Kamath**
Google Research
`pritish@alum.mit.edu`

**Dylan J. Foster**
Microsoft Research
`dylanfoster@microsoft.com`

**Nathan Srebro**
Toyota Technological
Institute at Chicago
`nati@ttic.edu`

## Abstract

We provide new insights on eluder dimension, a complexity measure that has been extensively used to bound the regret of algorithms for online bandits and reinforcement learning with function approximation. First, we study the relationship between the eluder dimension for a function class and a generalized notion of *rank*, defined for any monotone "activation" $\sigma : \mathbb{R} \to \mathbb{R}$, which corresponds to the minimal dimension required to represent the class as a generalized linear model. It is known that when $\sigma$ has derivatives bounded away from $0$, $\sigma$-rank gives rise to an upper bound on eluder dimension for any function class; we show however that eluder dimension can be exponentially smaller than $\sigma$-rank. We also show that the condition on the derivative is necessary; namely, when $\sigma$ is the relu activation, the eluder dimension can be exponentially larger than $\sigma$-rank. For binary-valued function classes, we obtain a characterization of the eluder dimension in terms of star number and threshold dimension, quantities which are relevant in active learning and online learning respectively.

## 1 Introduction

Russo and Van Roy [37] introduced the notion of **eluder dimension** for a function class and used it to analyze algorithms (based on the *Upper Confidence Bound (UCB)* and *Thompson Sampling* paradigms) for the multi-armed bandit problem with function approximation. Since then, eluder dimension has been extensively used to construct and analyze the regret of algorithms for contextual bandits and reinforcement learning (RL) with function approximation [see, e.g., 46, 36, 44, 6, 14, 19, 12, 16, 26, 24]. Even though the eluder dimension has become a central technique for reinforcement learning theory, little is known about when exactly it is bounded. This paper makes progress toward filling this gap in our knowledge.

Russo and Van Roy [37] established upper bounds on eluder dimension for (i) function classes for which inputs have finite cardinality (the "tabular" setting), (ii) linear functions over $\mathbb{R}^d$ of bounded norm, and (iii) generalized linear functions over $\mathbb{R}^d$ of bounded norm, with any activation that has derivatives bounded away from $0$. Apart from these function classes (and those that can be embedded into these), understanding of eluder dimension has been limited. Indeed, one might wonder whether a function class has "small" eluder dimension only if it can be realized as a class of (generalized) linear functions! This leads us to our first motivating question:

**Question 1.** *Are all function classes with small eluder dimension essentially generalized linear models?*

36th Conference on Neural Information Processing Systems (NeurIPS 2022).

Answering this question has substantial ramifications on the scope of prior work. An answer of "yes" would imply that the results in the aforementioned work which gives regret guarantees in terms of eluder dimension do not go beyond already-established regret guarantees for generalized linear bandit or RL settings [see, e.g. 17, 32, 45, 31]. An answer of "no" can be construed as a positive result for RL theory, as it would indicate that existing (and future) results which use the eluder dimension apply to a richer set of function classes than generalized linear models.

To answer Question 1, we first formally define what it means for a function class to be written as a generalized linear model (GLM). Informally, for an activation $\sigma : \mathbb{R} \to \mathbb{R}$ and a function class $\mathcal{F} \subseteq (\mathcal{X} \to \mathbb{R})$, we define the $\boldsymbol{\sigma}$-**rank** to be the smallest dimension $d$ needed to express every function in $\mathcal{F}$ as a generalized linear function in $\mathbb{R}^d$ with activation $\sigma$ (see Definition 3). Intuitively, the $\sigma$-rank captures the best possible upper bound on eluder dimension that the results from Russo and Van Roy [37] can give for a given $\mathcal{F}$ by treating it as a GLM with activation $\sigma$. We ask how the eluder dimension of any function class relates to its $\sigma$-rank for various activations $\sigma$. We show that the answer to Question 1 is indeed "no", i.e., there exists a function class with eluder dimension $d$ but the $\sigma$-rank *for any* monotone $\sigma$ is at least $\exp(\Omega(d))$. Thus, while Russo and Van Roy [37] show that the set of function classes with small eluder dimension is a *superset* of the set of GLM function classes, we (roughly speaking) show that the set of function classes with small eluder dimension is *strictly larger* than the set of GLM function classes.

We also prove that the requirement from Russo and Van Roy [37] that the derivative of the activation is bounded away from $0$ is necessary in order to bound the eluder dimension of GLMs. The upper bound in the paper [37] becomes vacuous when the activation function has zero derivative; we show a lower bound which indicates this requirement cannot be dropped. Namely, when $\sigma$ is the relu activation, we show that eluder dimension can be *exponentially larger* than $\sigma$-rank.

In a second line of inquiry, we study a *combinatorial* version of eluder dimension. The original definition of Russo and Van Roy [37] is defined for real-valued function classes, but one can specialize the definition to binary-valued function classes, leading to a so-called **combinatorial eluder dimension**. Thus, our second motivating question is:

**Question 2.** *Can we bound the combinatorial eluder dimension, perhaps in terms of more familiar learning-theoretic quantities?*

One might wonder: if the combinatorial eluder dimension is just a special case of the scale-sensitive version, why study it at all? Our reasons are threefold.

- The first and most immediate reason is that we are able to show new characterizations of eluder dimension once we move to the combinatorial definition. We elucidate a fundamental connection between eluder dimension and two other well-studied learning-theoretic quantities: (1) star number, a quantity that characterizes the label complexity of *pool-based active learning* [21], (2) threshold dimension, a quantity that characterizes the regret of *online learning* [3].[1] We believe that this new result may help us better understand how different learning tasks relate to each other.

- The second reason is that the combinatorial eluder dimension (or a multi-class variant of it) has already been studied for policy-based learning for contextual bandits and RL [see, e.g., 19, 34]. Thus, understanding the combinatorial eluder dimension may shed light on the challenges of policy-based RL.

- Our last reason has more philosophical bent. Historically, the discovery of VC dimension placed statistical learning theory on solid footing. Specifically, the fundamental theorem of statistical learning allows us to precisely characterize the statistical complexity of PAC learnability in terms of the combinatorial VC dimension. The insights from understanding the role of VC dimension in classification have led researchers to develop *scale-sensitive* complexity measures such as fat shattering dimension to provide sharper guarantees on learning. While we do not claim that the eluder dimension is fundamental to online RL and bandit settings, we believe that a better combinatorial understanding can lead to a better understanding of online decision making. In some sense, we are "working backwards" from the original scale-sensitive definition of eluder dimension to understand its combinatorial properties.

---

[1]Finiteness of the threshold dimension is equivalent to finiteness of Littlestone dimension [40, 22, 3].

## 1.1 Main contributions

In this work, we provide several results which show when eluder dimension can be bounded (or is unbounded). Our results can be separated into two categories: (1) those pertaining to the (scale-sensitive) eluder dimension (as originally defined by Russo and Van Roy [37]); (2) those pertaining to the combinatorial eluder dimension, defined for binary-valued function classes.

First, we investigate the relationship between eluder dimension and our notion of *generalized rank* that captures the realizability of any function class as a GLM. In Section 2, we formally introduce the eluder dimension (as well as a related quantity of the scale-sensitive star number). In Section 3, we formalize the notion of "generalized rank". In Section 4, we provide several results.

1. We show that eluder dimension can be *exponentially smaller* than $\sigma$-rank for any monotone activation $\sigma$, not just those with derivatives bounded away from 0 (Theorem 6).

2. We show that the condition on the derivative being bounded away from 0 is necessary for $\sigma$-rank to be an upper bound on eluder dimension. Namely, when $\sigma$ is the relu activation, we show that eluder dimension can be *exponentially larger* than $\sigma$-rank (Theorem 7).[2]

In Section 5, we specialize the eluder dimension to the binary-valued setting and present our results on the combinatorial eluder dimension.

1. We show that eluder dimension is finite if and only if both star number and threshold dimension are finite. Specifically, in Theorem 8 we show the following:

$$\max\{\mathsf{Sdim}(\mathcal{F}), \mathsf{Tdim}(\mathcal{F})\} \leq \mathsf{Edim}(\mathcal{F}) \leq \exp(O(\max\{\mathsf{Sdim}(\mathcal{F}), \mathsf{Tdim}(\mathcal{F})\})).$$

Furthermore, we demonstrate that both inequalities can be tight (see Theorem 9 and discussion above it).

2. We investigate the comparison between eluder dimension and sign-rk and prove stronger separations than in the scale-sensitive case; namely we show examples where one quantity is finite while the other is infinite (Theorem 10).

## 1.2 Related work

In this section, we highlight several related works.

**Bounds on eluder dimension.** Several papers provide bounds on eluder dimension for various function classes. The original bounds on tabular, linear, and generalized linear functions were proved by Russo and Van Roy [37] (and later generalized by Osband and Van Roy [36]). Mou et al. [34] provide several bounds for the combinatorial eluder dimension, mostly focusing on linear function classes. When the function class lies in an RKHS, Huang et al. [25] show that the eluder dimension is equivalent to the notion of *information gain* [41, 38], which can be seen as an infinite dimensional generalization of the fact that the eluder dimension for linear functions over $\mathbb{R}^d$ is $\tilde{\Theta}(d)$. In concurrent work, Dong et al. [12] also prove an exponential lower bound on the eluder dimension for ReLU networks.

**Applications of eluder dimension.** The main application of the eluder dimension is to design algorithms and prove regret guarantees for contextual bandits and reinforcement learning. A few examples include the papers [46, 36, 44, 6, 14, 19, 28, 16, 26, 24, 34]. While the majority of papers prove upper bounds via eluder dimension, Foster et al. [19] provided lower bounds for contextual bandits in terms of eluder dimension, if one is hoping for instance-dependent regret bounds. In addition, several works observe that eluder dimension sometimes does not characterize the sample complexity, as the guarantee via eluder dimension can be too loose [23, 20]. Beyond the online RL setting, eluder dimension has been applied to risk sensitive RL [15], Markov games [24, 29], representation learning [47], and active online learning [10].

**Other complexity measures for RL.** We also touch upon other complexity measures which have been suggested for RL. One category is Bellman/Witness rank approach [see e.g. 27, 11, 42], which is generalized to bilinear classes [13]. These complexity measures capture an interplay between

---

[2]This result was independently established by Dong et al. [12].

the MDP dynamics and the function approximator class; in contrast, eluder dimension is purely a property of the function approximator class and can be stated (and studied) without referring to an online RL problem. Jin et al. [28] define a *Bellman-Eluder dimension* which captures function classes which have small Bellman rank or eluder dimension. Lastly, Foster et al. [20] propose a *Decision-Estimation Coefficient* and prove that it is necessary and sufficient for sample-efficient interactive learning.

**Notions of rank.** The notion of rank we propose is a generalization of the classical notion of *sign rank*, also called *dimension complexity*. Sign rank has been studied extensively in combinatorics, learning theory, and communication complexity [see e.g. 1, 18, 5, 2, and references therein]. The norm requirements in our definition of $\sigma$-rank are related to the notion of *margin complexity* [5, 7, 30].

## 2 Eluder dimension and star number

Eluder dimension is a "sequential" notion of complexity for function classes, originally defined by Russo and Van Roy [37]. Informally speaking, it characterizes the longest sequence of *adversarially chosen* points one must observe in order to accurately estimate the function value at any other point. We consider a variant of the original definition, proposed by Foster et al. [19], that is never larger and is sufficient to analyze all the applications of eluder dimension in literature.

**Definition 1.** *For any function class $\mathcal{F} \subseteq (\mathcal{X} \to \mathbb{R})$, $f^\star \in \mathcal{F}$, and scale $\varepsilon \geq 0$, the **exact eluder dimension** $\underline{\mathsf{Edim}}_{f^\star}(\mathcal{F}, \varepsilon)$ is the largest $m$ such that there exists $(x_1, f_1), \ldots, (x_m, f_m) \in \mathcal{X} \times \mathcal{F}$ satisfying for all $i \in [m]$:*

$$|f_i(x_i) - f^\star(x_i)| > \varepsilon, \quad and \quad \sum_{j < i} (f_i(x_j) - f^\star(x_j))^2 \leq \varepsilon^2. \tag{1}$$

*Then for all $\varepsilon > 0$:*

- *the **eluder dimension** is $\mathsf{Edim}_{f^\star}(\mathcal{F}, \varepsilon) = \sup_{\varepsilon' \geq \varepsilon} \underline{\mathsf{Edim}}_{f^\star}(\mathcal{F}, \varepsilon')$.*

- *$\underline{\mathsf{Edim}}(\mathcal{F}, \varepsilon) := \sup_{f^\star \in \mathcal{F}} \underline{\mathsf{Edim}}_{f^\star}(\mathcal{F}, \varepsilon)$ and $\mathsf{Edim}(\mathcal{F}, \varepsilon) := \sup_{f^\star \in \mathcal{F}} \mathsf{Edim}_{f^\star}(\mathcal{F}, \varepsilon)$.*

This definition is never larger than the original definition of Russo and Van Roy [37], which asks for a witnessing *pair* of functions $f_i, f_i' \in \mathcal{F}$ (the above restricts $f_i' = f^\star$). Hence, all lower bounds on our variant of eluder dimension immediately apply to the original definition. Moreover, all upper bounds on eluder dimension in this paper can also be shown to hold for the other definition (unless stated otherwise).

We also consider the closely related notion of *star number* defined by Foster et al. [19], which generalizes a combinatorial parameter introduced in the active learning literature by Hanneke and Yang [21] (we will denote it as Sdim for consistency). We study the combinatorial star number in more detail in Section 5. The *only* difference between the definitions of eluder dimension and star number is that $\sum_{j < i}$ is replaced by $\sum_{j \neq i}$, which makes the star number a "non-sequential" notion of complexity.

**Definition 2.** *For any function class $\mathcal{F} \subseteq (\mathcal{X} \to \mathbb{R})$, $f^\star \in \mathcal{F}$, and scale $\varepsilon \geq 0$, the **exact star number** $\underline{\mathsf{Sdim}}_{f^\star}(\mathcal{F}, \varepsilon)$ is the largest $m$ such that there exists $(x_1, f_1), \ldots, (x_m, f_m) \in \mathcal{X} \times \mathcal{F}$ satisfying for all $i \in [m]$:*

$$|f_i(x_i) - f^\star(x_i)| > \varepsilon, \quad and \quad \sum_{j \neq i} (f_i(x_j) - f^\star(x_j))^2 \leq \varepsilon^2.$$

*Then for all $\varepsilon > 0$:*

- *the **star number** is $\mathsf{Sdim}_{f^\star}(\mathcal{F}, \varepsilon) = \sup_{\varepsilon' \geq \varepsilon} \underline{\mathsf{Sdim}}_{f^\star}(\mathcal{F}, \varepsilon')$.*

- *$\underline{\mathsf{Sdim}}(\mathcal{F}, \varepsilon) := \sup_{f^\star \in \mathcal{F}} \underline{\mathsf{Sdim}}_{f^\star}(\mathcal{F}, \varepsilon)$ and $\mathsf{Sdim}(\mathcal{F}, \varepsilon) := \sup_{f^\star \in \mathcal{F}} \mathsf{Sdim}_{f^\star}(\mathcal{F}, \varepsilon)$.*

It immediately follows from these definitions that the star number is never larger than eluder dimension. On the other hand, the star number can be arbitrarily smaller than eluder dimension.

**Proposition 1.** *For all $\mathcal{F}$, $f^* \in \mathcal{F}$ and scale $\varepsilon \geq 0$, it holds that*[3]

$$\underline{\mathsf{Sdim}}_{f^*}(\mathcal{F}, \varepsilon) \;\leq\; \underline{\mathsf{Edim}}_{f^*}(\mathcal{F}, \varepsilon) \;\leq\; \min\{|\mathcal{X}|, |\mathcal{F}| - 1\} \,.$$

**Proposition 2** (simplified from Foster et al. [19, Prop 2.3])**.** *For the class of threshold functions given as $\mathcal{F}_n^{\mathrm{th}} := \{f_i : [n] \to \{0,1\} \mid i \in [n+1]\}$, where $f_i(x) := \mathbb{1}\{x \geq i\}$, and for $f^\star = f_{n+1}$, it holds for all $\varepsilon < 1$ that $\underline{\mathsf{Sdim}}_{f^\star}(\mathcal{F}, \varepsilon) = 2$ and $\underline{\mathsf{Edim}}_{f^\star}(\mathcal{F}, \varepsilon) = n$.*

## 3 Generalized rank

*Dimension complexity* has been studied extensively in combinatorics, learning theory, and communication complexity [see e.g. 1, 18, 5, 2]. The classical notion of dimension complexity, also known as *sign rank*, corresponds to the smallest dimension required to embed the input space such that all hypotheses in the function class under consideration are realizable as halfspaces. We consider a generalized notion of rank that is specified for any particular activation $\sigma : \mathbb{R} \to \mathbb{R}$, and captures to the smallest dimension required to represent the function class as a GLM when $\sigma$ is the activation. In what follows, we let $\mathcal{B}_d(R) := \{x \in \mathbb{R}^d \mid \|x\|_2 \leq R\}$.

**Definition 3.** *For any $\sigma : \mathbb{R} \to \mathbb{R}$, the $\boldsymbol{\sigma}$-rank of a function class $\mathcal{F} \subseteq (\mathcal{X} \to \mathbb{R})$ at scale $R > 0$, denoted as $\sigma\text{-}\mathsf{rk}(\mathcal{F}, R)$, is the smallest dimension $d$ for which there exists mappings $\phi : \mathcal{X} \to \mathcal{B}_d(1)$ and $w : \mathcal{F} \to \mathcal{B}_d(R)$ such that*[4]

$$\text{for all } (x, f) \in \mathcal{X} \times \mathcal{F} \;:\; f(x) \;=\; \sigma(\langle w(f), \phi(x) \rangle), \tag{2}$$

*or $\infty$ if no such $d$ exists. For a collection of activation functions $\Sigma \subseteq (\mathbb{R} \to \mathbb{R})$, the $\boldsymbol{\Sigma}$-rank is*

$$\Sigma\text{-}\mathsf{rk}(\mathcal{F}, R) \;:=\; \min_{\sigma \in \Sigma} \sigma\text{-}\mathsf{rk}(\mathcal{F}, R).$$

**Examples.** We present some examples of $\Sigma$-rk that motivate our definition above.

1. **Threshold activation.** $\mathsf{sign}(z)$ yields the classic notion of *sign-rank* (equivalent to *dimension complexity*, as already mentioned). In this case, the scale $R$ is irrelevant, so we denote $\mathsf{sign}\text{-}\mathsf{rk}(\mathcal{F}) := \mathsf{sign}\text{-}\mathsf{rk}(\mathcal{F}, R)$ for any $R$. Note that this quantity is meaningful only for $\mathcal{F} \subseteq (\mathcal{X} \to \{-1, 1\})$.

2. **Identity activation.** For $\mathsf{id}(z) := z$, $\mathsf{id}\text{-}\mathsf{rk}(\mathcal{F}, R)$ is the smallest dimension needed to represent each $f \in \mathcal{F}$ as a (norm-bounded) linear function. We abbreviate $\mathsf{rk} := \mathsf{id}\text{-}\mathsf{rk}$, as this corresponds to the standard notion of rank of the matrix $(f(x))_{x,f}$ (albeit with the additional norm constraint).

3. **Monotone activations.** For $L \geq \mu \geq 0$, $\mathcal{M}_\mu^L$ consists of all activations $\sigma$ such that for all $z < z'$, it holds that $\mu \leq \frac{\sigma(z') - \sigma(z)}{z' - z} \leq L$ (for differentiable $\sigma$, this is equivalent to $\mu \leq \sigma'(z) \leq L$ for all $z \in \mathbb{R}$).[5] An important special case is when $\mu = 0$, and a particular $\sigma \in \mathcal{M}_0^1$ of interest is the rectified linear unit (ReLU) defined as $\mathsf{relu}(z) := \max\{z, 0\}$. For ease of notation, we denote $\mathcal{M}_\mu := \mathcal{M}_\mu^1$.

   While we will always be explicit about the Lipschitz constant, note that the scale of the Lipschitz constant $L$ (and $\mu$) is interchangable with the scale of $R$. In particular,

   $$\mathcal{M}_\mu^L\text{-}\mathsf{rk}(\mathcal{F}, R) \;=\; \mathcal{M}_{\mu/L}\text{-}\mathsf{rk}(\mathcal{F}, RL). \tag{3}$$

4. **All activations.** $\Sigma^{\mathrm{all}}$ consists of all activations $\sigma$. We mention this notion of rank only in passing, and we will not focus on it for the rest of the paper.

We present a result which relates the aforementioned quantities (proof in Appendix A).

---

[3]For the definition of eluder dimension considered by [37], an upper bound of $\min\{|\mathcal{X}|, \binom{|F|}{2}\}$ holds, which can be tight. This upper bound holds because the witnessing pair of functions $(f_i, f_i')$ has to be distinct for each $i$.

[4]Note that only the product of the scales of $\phi$ and $w$ is relevant. The definition remains equivalent if we let $\phi : \mathcal{X} \to \mathcal{B}_d(R_\phi)$ and $w : \mathcal{F} \to \mathcal{B}_d(R_w)$ for any $R_\phi$ and $R_w$ such that $R = R_\phi \cdot R_w$.

[5]To prove upper bounds on eluder dimension, it suffices for this condition to hold only when $|z| \leq R$, [see e.g. 37]. Since we fix $\sigma$ in our definition first and then consider $\sigma$-rank at different scales $R$, this weaker condition complicates our definitions. Note that at any specific scale $R$, we can always modify $\sigma$ to satisfy the required constraint everywhere by extending it linearly whenever $|z| > R$.

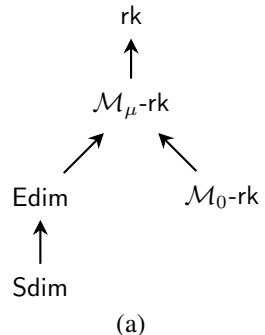

| | rk | $\mathcal{M}_\mu$-rk | $\mathcal{M}_0$-rk | Edim | Sdim |
|---|---|---|---|---|---|
| rk | | | | | |
| $\mathcal{M}_\mu$-rk | $\mathcal{F}^{\mathrm{exp}}$ | | | | |
| $\mathcal{M}_0$-rk | $\mathcal{F}^{\mathrm{relu}}$ | | | $\mathcal{F}^{\mathrm{relu}}$ | |
| Edim | $\mathcal{F}^{\oplus}$ | | | | |
| Sdim | | | | $\mathcal{F}^{\mathrm{th}}$ | |

| (a) | (b) |
|---|---|

Figure 1: (a) Each arrow $M_1 \to M_2$ indicates that $M_1(\mathcal{F}) \lesssim M_2(\mathcal{F})$ for all $\mathcal{F}$, where the dependence on $R$ and $\varepsilon$ is suppressed for clarity (see Propositions 1, 3, 4 for precise bounds). Whenever $M_2 \to M_1$ arrow is missing, there is an example of a class $\mathcal{F}$ where $M_1(\mathcal{F}) \ll M_2(\mathcal{F})$. (b) An entry $\mathcal{F}$ in $(M_1, M_2)$ means that $M_1(\mathcal{F}) \ll M_2(\mathcal{F})$. Green cells indicate that $M_1(\mathcal{F}) \gtrsim M_2(\mathcal{F})$ for all $\mathcal{F}$.

**Proposition 3.** *For all $\mathcal{F} \subseteq (\mathcal{X} \to \mathbb{R})$, $R > 0$ and $\mu \in (0, 1]$, we have:*

$$\mathsf{rk}(\mathcal{F}, R) \;\geq\; \mathcal{M}_\mu\text{-}\mathsf{rk}(\mathcal{F}, R) \;\geq\; \mathcal{M}_0\text{-}\mathsf{rk}(\mathcal{F}, R) \;\geq\; \mathsf{sign\text{-}rk}(\mathcal{F}) - 1,$$

*where the last inequality is meaningful only for $\mathcal{F} \subseteq (\mathcal{X} \to \{-1, 1\})$. Moreover, for each inequality above, there exists a function class $\mathcal{F}$ which exhibits an infinite gap between the two quantities.*

## 4 Eluder dimension versus generalized rank

In this section, we compare eluder dimension and star number with each notion of generalized rank: rk, $\mathcal{M}_\mu$-rk (for $\mu > 0$) and $\mathcal{M}_0$-rk. Our results are summarized in Figure 1.

**Eluder vs. rk and $\mathcal{M}_\mu$-rk.** Russo and Van Roy [37] and Osband and Van Roy [36] provided upper bounds on eluder dimension for linear and generalized linear function classes. For completeness, we restate this result, with a slight improvement and include the proof with precise dependence on problem parameters in Appendix B. Intuitively, the generalized linear rank allows us to capture the tightest possible upper bound that the guarantees in the papers [37, 36] can provide on eluder dimension.

**Proposition 4** (cf. [37], Prop. 6, 7; [36], Prop. 2, 4)**.** *For any function class $\mathcal{F} \subseteq (\mathcal{X} \to \mathbb{R})$ and $\varepsilon > 0$:*

*(i) For all $R > 0$, $\underline{\mathsf{Edim}}(\mathcal{F}, \varepsilon) \;\leq\; \mathsf{rk}(\mathcal{F}, R) \cdot O\left(\log \frac{R}{\varepsilon}\right)$.*

*(ii) For all $L, \mu, R > 0$, $\underline{\mathsf{Edim}}(\mathcal{F}, \varepsilon) \;\leq\; \mathcal{M}_\mu^L\text{-}\mathsf{rk}(\mathcal{F}, R) \cdot O\left(\frac{L^2}{\mu^2} \log\left(\frac{RL}{\varepsilon}\right)\right)$.*

This result has been used to prove upper bounds on eluder dimension of various function classes beyond GLMs; for example, the class of bounded degree polynomials, by taking the feature map $\phi(x)$ to be the vector of low degree monomials. The upper bound in Part (i) is in fact tight (up to constants) for the class of linear functions, as shown in Proposition 5 below. This trivially implies the optimality of the bound in Part (ii) up to the factor of $(L/\mu)^2$ which to the best of our knowledge is open.

**Proposition 5** ([33])**.** *For any $R > 0$, $\mathcal{X} := \mathcal{B}_d(1)$ and $\mathcal{F} := \{f_\theta : x \mapsto \langle \theta, x \rangle \mid \theta \in \mathcal{B}_d(R)\}$, it holds that:*

$$\underline{\mathsf{Edim}}(\mathcal{F}, \varepsilon) \;\geq\; \Omega\left(d \log\left(\frac{R}{\varepsilon}\right)\right) .$$

For completeness, we include the proof in Appendix C.

**Eluder vs. $\mathcal{M}_0$-rk.** It turns out that eluder dimension and $\mathcal{M}_0$-rk are incomparable. That is, there exists a function class for which eluder dimension is exponentially smaller than $\mathcal{M}_0$-rk (and hence $\mathcal{M}_\mu$-rk and rk by Proposition 3). Moreover, there exists a different function class for which eluder dimension (even the star number) is exponentially larger than relu-rk (and hence $\mathcal{M}_0$-rk).

First, we show that the eluder dimension can be exponentially smaller than $\mathcal{M}_0$-rk for the class of parities over $d$ bits. Thus, parities over $d$ bits exhibits an example where the eluder dimension is exponentially smaller than the best possible bound one can derive using the existing results of Proposition 4.

**Theorem 6.** *For* $\mathcal{X} = \{-1,1\}^d$ *and* $\mathcal{F}^\oplus := \left\{ f_S : x \mapsto \prod_{i \in S} x_i \mid S \subseteq [d] \right\}$, *it holds that*

(i) $\mathcal{M}_0$-$\mathsf{rk}(\mathcal{F}^\oplus, R) \geq 2^{d/2} - 1$ *for all* $R > 0$.

(ii) $\underline{\mathsf{Sdim}}(\mathcal{F}^\oplus, \varepsilon) \leq \underline{\mathsf{Edim}}(\mathcal{F}^\oplus, \varepsilon) \leq d$ *for all* $\varepsilon \geq 0$.

*Proof.* Part (i). From Proposition 3, we have that $\mathcal{M}_0$-$\mathsf{rk}(\mathcal{F}^\oplus, R) \geq \mathsf{sign}$-$\mathsf{rk}(\mathcal{F}^\oplus) - 1$ for any $\sigma \in \mathcal{M}_0$. The proof is now complete by noting a well known result that $\mathsf{sign}$-$\mathsf{rk}(\mathcal{F}^\oplus) \geq 2^{d/2}$ [18].

Part (ii). For any $x \in \{-1,1\}^d$ consider its 0-1 representation $\widetilde{x} \in \mathbb{F}_2^d$ (representing $+1$ by 0 and $-1$ by 1). All functions in $\mathcal{F}^\oplus$ can be simply viewed as linear functions over $\mathbb{F}_2$. Namely, any parity function is indexed by a vector $a \in \mathbb{F}_2^d$, with $f_a(x) := (-1)^{\langle a, \widetilde{x} \rangle}$.

Note that $\underline{\mathsf{Edim}}(\mathcal{F}^\oplus, \varepsilon) = 0$ for all $\varepsilon \geq 2$. For any $\varepsilon < 2$, suppose $\underline{\mathsf{Edim}}_{f^\star}(\mathcal{F}^\oplus, \varepsilon) = m$, witnessed by $(x_1, f_{a_1}), \ldots, (x_m, f_{a_m}) \in \{-1,1\}^d$ and $f^\star = f_{a^\star}$. We have

- $f_{a_i}(x_i) \neq f_{a^\star}(x_i)$, and

- $f_{a_i}(x_j) = f_{a^\star}(x_j)$ for all $j < i$ : since $\sum_{j<i} (f_{a_i}(x_j) - f_{a^\star}(x_j))^2 < \varepsilon^2 < 4$ iff all terms are 0.

Thus, we have $\langle a_i - a^\star, \widetilde{x} \rangle = 0$ for all $\widetilde{x} \in \mathbb{F}_2$-$\mathsf{span}(\{\widetilde{x}_1, \ldots, \widetilde{x}_{i-1}\})$. But $\langle a_i - a^\star, \widetilde{x}_i \rangle = 1$ and hence $\widetilde{x}_i$ is linearly independent of $\{\widetilde{x}_1, \ldots, \widetilde{x}_{i-1}\}$ over $\mathbb{F}_2^d$. Thus, $\{\widetilde{x}_1, \ldots, \widetilde{x}_m\}$ are all linearly independent over $\mathbb{F}_2^d$, and hence $m \leq d$. $\qquad\square$

The bound in part (ii) of Theorem 6 was also calculated by Mou et al. [34, Prop. 3].

Next, we show a separation in the other direction for eluder dimension vs. $\mathcal{M}_0$-rk using the ReLU function class. Thus, we cannot hope to remove the requirement for the activation function $\sigma$ to be strictly monotonically increasing in Proposition 4 (ii) for bounding the eluder dimension.

**Theorem 7.** *Let* $R > 0$ *and* $\mathcal{X} = \mathcal{B}_d(1)$. *Define*

$$\mathcal{F}^{\mathrm{relu}} := \left\{ f_{\theta, b} : x \mapsto \mathsf{relu}(\langle \theta, x \rangle + b) \mid \|\theta\|^2 + b^2 \leq R^2 \right\}.$$

*It holds that*

(i) $\mathcal{M}_0$-$\mathsf{rk}(\mathcal{F}^{\mathrm{relu}}, R) \leq \mathsf{relu}$-$\mathsf{rk}(\mathcal{F}^{\mathrm{relu}}, R) \leq d + 1$,

(ii) $\underline{\mathsf{Edim}}(\mathcal{F}^{\mathrm{relu}}, \varepsilon) \geq \underline{\mathsf{Sdim}}(\mathcal{F}^{\mathrm{relu}}, \varepsilon) \geq \left(\frac{R}{4\varepsilon}\right)^{d/2}$ *for all* $\varepsilon \in (0, \frac{R}{4})$.

*Proof.* Part (i) is immediate from the definition. We show Part (ii) in the special case of $R = 2$; the general case follows by relatively scaling $\varepsilon$, since relu is *homogeneous*, namely, $\mathsf{relu}(ax) = a \cdot \mathsf{relu}(x)$. Consider any $U \subseteq \mathcal{X}$ such that $\|u\| = 1$ and $\langle u, v \rangle \leq 1 - \varepsilon$ for all $u, v \in U$. It holds that $\underline{\mathsf{Sdim}}_{f^\star}(\mathcal{F}^{\mathrm{relu}}, \varepsilon) \geq |U|$ when $f^\star$ is the identically zero function, since the function $f_u(x) = \mathsf{relu}(\langle u, x \rangle - (1 - \varepsilon))$ is such that $f_u(v) = 0$ for all $v \in U \setminus \{u\}$, whereas $f_u(u) = \varepsilon$. A standard sphere packing argument shows that such a set $U$ exists with $|U| \geq (1/2\varepsilon)^{d/2}$ for all $\varepsilon < 1/2$. In particular, the $\delta$-packing number of the unit sphere is at least $(1/\delta)^d$ [43, Cor. 4.2.13]. Thus, we can find $(1/\delta)^d$ points such that each pair $u, v$ satisfies $\|u - v\| \geq \delta$, or equivalently $\langle u, v \rangle \leq 1 - \delta^2/2$. Setting $\delta = \sqrt{2\varepsilon}$ proves the claimed lower bound. $\qquad\square$

Theorem 7 was independently shown by Dong et al. [12, Thm. 5.1].

We remark that while we considered the variant of eluder dimension as defined by Foster et al. [19], the lower bound on eluder dimension in Theorem 7 immediately holds for the notion defined by Russo and Van Roy [37]. On the other hand, the upper bound on eluder dimension in Theorem 6 can be shown to hold even with the definition of Russo and Van Roy [37] (by replacing every instance of $a^\star$ by $a_i'$).

| | $f_1$ | $f_2$ | $f_3$ | $f_4$ | $f_5$ | $f_6$ |
|---|---|---|---|---|---|---|
| $x_1$ | 1 | 0 | 0 | 0 | 0 | 0 |
| $x_2$ | * | 1 | 0 | 0 | 0 | 0 |
| $x_3$ | * | * | 1 | 0 | 0 | 0 |
| $x_4$ | * | * | * | 1 | 0 | 0 |
| $x_5$ | * | * | * | * | 1 | 0 |
| $x_6$ | * | * | * | * | * | 1 |

Eluder sequence

| | $f_1$ | $f_2$ | $f_3$ | $f_4$ | $f_5$ | $f_6$ |
|---|---|---|---|---|---|---|
| $x_1$ | 1 | 0 | 0 | 0 | 0 | 0 |
| $x_2$ | 0 | 1 | 0 | 0 | 0 | 0 |
| $x_3$ | 0 | 0 | 1 | 0 | 0 | 0 |
| $x_4$ | 0 | 0 | 0 | 1 | 0 | 0 |
| $x_5$ | 0 | 0 | 0 | 0 | 1 | 0 |
| $x_6$ | 0 | 0 | 0 | 0 | 0 | 1 |

Star sequence

| | $f_1$ | $f_2$ | $f_3$ | $f_4$ | $f_5$ | $f_6$ |
|---|---|---|---|---|---|---|
| $x_1$ | 1 | 0 | 0 | 0 | 0 | 0 |
| $x_2$ | 1 | 1 | 0 | 0 | 0 | 0 |
| $x_3$ | 1 | 1 | 1 | 0 | 0 | 0 |
| $x_4$ | 1 | 1 | 1 | 1 | 0 | 0 |
| $x_5$ | 1 | 1 | 1 | 1 | 1 | 0 |
| $x_6$ | 1 | 1 | 1 | 1 | 1 | 1 |

Threshold sequence

Figure 2: Illustration of witnessing sequences of length 6 for eluder dimension, star number and threshold dimension with respect to $f^\star = 0$ (we use the 0/1 representation of functions for clarity). '*' in the eluder witness sequence refers to a free value, either 0 or 1.

## 5 Relationships between combinatorial measures

In this section, we specialize the eluder dimension to binary-valued function classes and prove several additional characterizations that relate this combinatorial eluder dimension to other learning-theoretic quantities. First, we (re)define these learning-theoretic quantities. The first two definitions (of combinatorial eluder dimension and star number respectively) are the specialization of the scale-sensitive versions (cf. Definition 1 and 2) to the binary-valued function setting. We abuse notation by dropping the argument $\varepsilon$ from previous definitions in order to be consistent.

**Definition 4.** *Fix any function class $\mathcal{F} \subseteq (\mathcal{X} \to \{1, -1\})$ and any $f^\star \in \mathcal{F}$.*

- *The **combinatorial eluder dimension** w.r.t. $f^\star$, denoted $\mathsf{Edim}_{f^\star}(\mathcal{F})$, is defined as the largest $m$ such that there exists $(x_1, f_1), \ldots, (x_m, f_m) \in \mathcal{X} \times \mathcal{F}$ satisfying for all $i \in [m]$:*

$$f_i(x_i) \neq f^\star(x_i), \quad \text{and} \quad \text{for all } j < i: \quad f_i(x_j) = f^\star(x_j).$$

- *The **star number** w.r.t. $f^\star$, denoted $\mathsf{Sdim}_{f^\star}(\mathcal{F})$, is defined as the largest $m$ such that there exists $(x_1, f_1), \ldots, (x_m, f_m) \in \mathcal{X} \times \mathcal{F}$ satisfying for all $i \in [m]$:*

$$f_i(x_i) \neq f^\star(x_i), \quad \text{and} \quad \text{for all } j \neq i: \quad f_i(x_j) = f^\star(x_j).$$

- *The **threshold dimension** w.r.t. $f^\star$, denoted $\mathsf{Tdim}_{f^\star}(\mathcal{F})$, is defined as the largest $m$ such that there exists $(x_1, f_1), \ldots, (x_m, f_m) \in \mathcal{X} \times \mathcal{F}$ satisfying for all $i \in [m]$:*

$$\text{for all } k \geq i: \quad f_i(x_k) \neq f^\star(x_k), \quad \text{and} \quad \text{for all } j < i: \quad f_i(x_j) = f^\star(x_j).$$

*As before, we define the combinatorial eluder dimension (resp. star number and threshold dimension) to be $\mathsf{Edim}(\mathcal{F}) := \sup_{f \in \mathcal{F}} \mathsf{Edim}_f(\mathcal{F})$ ($\mathsf{Sdim}(\mathcal{F})$ and $\mathsf{Tdim}(\mathcal{F})$ are defined similarly).*

Let us pause to unpack these definitions and give some background. In fact, the star number definition stated above is the original definition of Hanneke and Yang [21], who give tight upper and lower bounds on the label complexity of *pool-based active learning* via the star number $\mathsf{Sdim}(\mathcal{F})$ and show that almost every previously proposed complexity measure for active learning takes a worst case value equal to the star number. Roughly speaking, the star number corresponds to the number of "singletons" one can embed in a function class; that is, the maximum number of functions that differ from a base function $f^\star$ at exactly one point among a subset of the domain $\{x_1, \ldots, x_m\} \subseteq \mathcal{X}$.

The threshold dimension has recently gained attention due to its role in proving an equivalence relationship between private PAC learning and online learning [see, e.g., 3, 9]. We slightly generalize the definition of Alon et al. [3] to allow for any base function $f^\star$, in the spirit of the other two definitions. A classical result in model theory provides a link between the threshold dimension and Littlestone dimension (which we denote $\mathsf{Ldim}$), a quantity which is both necessary and sufficient for online learnability [8, 4]. In particular, results by Shelah [40] and Hodges et al. [22] show that for any binary-valued $\mathcal{F}$ and any $f^\star \in \mathcal{F}$:

$$\lfloor \log \mathsf{Tdim}_{f^\star}(\mathcal{F}) \rfloor \leq \mathsf{Ldim} \leq 2^{\mathsf{Tdim}_{f^\star}(\mathcal{F})}.$$

A combinatorial proof of this fact can be found in Thm. 3 of Alon et al. [3].[6] Thus, finiteness of threshold dimension is necessary and sufficient for online learnability (albeit in a much weaker, "qualitative" sense).

## 5.1 A qualitative equivalence

Now, we prove a rather surprising characterization that closely ties all three quantities in Definition 4 together. Our result implies that for any binary-valued function class, finiteness of the combinatorial eluder dimension is *equivalent* to finiteness of both star number and threshold dimension.

**Theorem 8.** *For any function class $\mathcal{F} \subseteq (\mathcal{X} \to \{1, -1\})$ and any $f^\star \in \mathcal{F}$, the following holds:*

$$\max\{\mathsf{Sdim}_{f^\star}(\mathcal{F}), \mathsf{Tdim}_{f^\star}(\mathcal{F})\} \leq \mathsf{Edim}_{f^\star}(\mathcal{F}) \leq 4^{\max\{\mathsf{Sdim}_{f^\star}(\mathcal{F}), \mathsf{Tdim}_{f^\star}(\mathcal{F})\}}.$$

The proof of Theorem 8 can be found in Appendix D. The lower bound is trivial by examining Definition 4. To prove the upper bound, we rely on a novel connection to Ramsey theory. In particular, we show that sequences $(x_1, f_1), \ldots (x_m, f_m)$ which witness $\mathsf{Edim}_{f^\star} = m$ form a bijection with red-blue colorings of the complete graph $K_m$, while subsequences of the witnessing eluder sequence that are valid star number witnesses or threshold dimension witnesses can be interpreted as monochromatic colorings of subgraphs of $K_m$ (see Figure 2). Thus, applying classical bounds from Ramsey theory implies the result.

Theorem 8 has an exponential gap between the upper and lower bounds. Can we improve either of the inequalities? The lower bound cannot be improved, by considering the simple examples over $\mathcal{X} = [n]$ of the singleton class $\mathcal{F}_n^{\mathrm{sing}} := \{x \mapsto \mathbb{1}\{x = i\} \mid i \in [n+1]\}$ and the threshold class $\mathcal{F}_n^{\mathrm{th}} := \{x \mapsto \mathbb{1}\{x \geq i\} \mid i \in [n+1]\}$. We also show that the upper bound cannot be improved, for example, to $\mathsf{Edim}(\mathcal{F}) \leq \mathrm{poly}(\mathsf{Sdim}(\mathcal{F}), \mathsf{Tdim}(\mathcal{F}))$ in general.

**Theorem 9.** *For every $N > 0$, there exists a function class $\mathcal{F}_N$ such that $\mathsf{Edim}(\mathcal{F}_N) = N$ and $\max\{\mathsf{Sdim}(\mathcal{F}), \mathsf{Tdim}(\mathcal{F})\} < c \cdot \log_2 N$, where $c > 1/2$ is some absolute numerical constant.*

The proof of Theorem 9 can be found in Appendix E. It relies on the probabilistic method to show the existence of a randomly constructed $\mathcal{F}$ which satisfies the desired properties.

## 5.2 Comparisons with sign-rk

In this section, we investigate the comparison of these combinatorial quantities (eluder, star, threshold) with sign-rk, and examine whether we can prove stronger separations. One direction is clear; for the class of linear classifiers in $\mathbb{R}^d$, the combinatorial eluder dimension, star number, and threshold dimension are all infinite. However, we ask if the other separation is also possible: can we construct $\mathcal{F}$ where eluder/star/threshold are finite but sign-rk $= \infty$? We have already provided an explicit exponential separation: Theorem 6 shows that for the function class $\mathcal{F}^\oplus$, we had $\mathsf{Edim}(\mathcal{F}^\oplus) = \mathsf{Sdim}(\mathcal{F}^\oplus) = d$ but sign-rk$(\mathcal{F}^\oplus) \geq 2^{d/2}$. (One can also show that $\mathsf{Tdim}(\mathcal{F}^\oplus) = d$.) We are able to show stronger (but nonconstructive) separations by extending the probabilistic techniques of Alon et al. [2], who recently provided similar separations for VC dimension versus sign-rk. In view of Proposition 3, this result also provides a separation for the scale-sensitive Definition 1.

**Theorem 10.** *For every $N > 0$, there exists a function class $\mathcal{F}_N \subseteq ([N] \to \{1, -1\})$ such that $\mathsf{Edim}_1(\mathcal{F}_N) = 4$ and sign-rk$(\mathcal{F}_N) \geq \Omega(N^{1/9}/\log N)$, where $1$ is shorthand for the all 1s function.*

The proof of Theorem 10 can be found in Appendix F. It is straightforward to replace the reference function $f^\star(x) = 1$ with any fixed reference function $f^\star : [N] \to \{1, -1\}$. First, we use Lemma 22 of Alon et al. [2] which bounds the number of distinct matrices with sign-rk $= r$; then using a probabilistic argument we show that there must be many (more) matrices with $\mathsf{Edim}_1 = 4$, so at least one of them must have large sign-rk.

The careful reader might notice that we do not prove the existence of a function class where $\mathsf{Edim}(\mathcal{F})$ is constant and the sign-rk is infinite; instead we prove the weaker statement that a function class exists with $\mathsf{Edim}$ w.r.t. any *fixed* function $f^\star$ is bounded. We conjecture that the stronger statement holds; see Appendix F.1 for more details.

---

[6]Alon et al. [3] prove the result for $f^\star(x) = -1$, but it is easy to extend their proof to hold for any $f^\star$.

### 5.3 Back to scale-sensitive?

It is natural to ask if our results can be extended back to the scale-sensitive definitions. First, we require a scale-sensitive version of threshold dimension. One proposal is the following:

**Definition 5.** *For any function class $\mathcal{F} \subseteq (\mathcal{X} \to \mathbb{R})$, $f^\star \in \mathcal{F}$, and scale $\varepsilon \geq 0$, the **exact threshold dimension** $\underline{\mathsf{Tdim}}_{f^\star}(\mathcal{F}, \varepsilon)$ is the largest $m$ such that there exists $(x_1, f_1), \ldots, (x_m, f_m) \in \mathcal{X} \times \mathcal{F}$ satisfying for all $i \in [m]$:*

$$\forall j \geq i \ : |f_i(x_j) - f^\star(x_j)| > \varepsilon, \quad and \quad \sum_{j < i} (f_i(x_j) - f^\star(x_j))^2 \leq \varepsilon^2.$$

*Then for all $\varepsilon > 0$:*

- *the **threshold dimension** is $\mathsf{Tdim}_{f^\star}(\mathcal{F}, \varepsilon) = \sup_{\varepsilon' \geq \varepsilon} \underline{\mathsf{Tdim}}_{f^\star}(\mathcal{F}, \varepsilon')$.*

- *$\underline{\mathsf{Tdim}}(\mathcal{F}, \varepsilon) := \sup_{f^\star \in \mathcal{F}} \underline{\mathsf{Tdim}}_{f^\star}(\mathcal{F}, \varepsilon)$ and $\mathsf{Tdim}(\mathcal{F}, \varepsilon) := \sup_{f^\star \in \mathcal{F}} \mathsf{Tdim}_{f^\star}(\mathcal{F}, \varepsilon)$.*

Definition 5 mirrors the scale-sensitive definitions for eluder and star; it also recovers the combinatorial definition when $\mathcal{F}$ is binary-valued. By definition, the relationship that $\mathsf{Edim}_{f^\star}(\mathcal{F}, \varepsilon) \geq \max\{\mathsf{Sdim}_{f^\star}(\mathcal{F}, \varepsilon), \mathsf{Tdim}_{f^\star}(\mathcal{F}, \varepsilon)\}$ for every $\mathcal{F}, f^\star \in \mathcal{F}, \varepsilon > 0$ is trivial. However, one cannot hope to prove the corresponding upper bound under this definition. For example, for any $\varepsilon > 0$, take the function class which is represented by the $N \times N$ matrix:

$$f_j(x_i) = \begin{cases} 0 & i < j \\ \varepsilon & i = j \\ 0.99\varepsilon & i > j. \end{cases}$$

It is easy to see that $\mathsf{Edim}_0(\mathcal{F}, \varepsilon) = N$, while $\mathsf{Sdim}_0(\mathcal{F}, \varepsilon) = 2$ and $\mathsf{Tdim}_0(\mathcal{F}, \varepsilon) = 1$. Notice that this class is still "threshold-like", but Definition 5 does not capture this for said value of $\varepsilon$. Generally speaking, it is unclear how to carry over the intuition from Ramsey theory that applies in the combinatorial case to the scale-sensitive case; we leave this to future work.

## Acknowledgments and Disclosure of Funding

We thank Gaurav Mahajan for allowing us to include the proof of Proposition 5 [33]. We thank Akshay Krishnamurthy, Tengyu Ma, and Ruosong Wang for helpful discussions. GL was partially supported by NSF award IIS-1764032. PK was partially supported by NSF BIGDATA award 1546500. Part of this work was done while GL, PK, and DF were participating in the Simons program on the Theoretical Foundations of Reinforcement Learning.

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
