# OpenReview forum: "Understanding the Eluder Dimension"
_NeurIPS.cc/2022/Conference — NeurIPS 2022 Accept_

### Official Review · Reviewer_kWVT · 2022-07-12

**Rating:** 5
**Confidence:** 3
**Soundness:** 3 good
**Presentation:** 2 fair
**Contribution:** 2 fair

**Summary:**

This paper provides new insights on the eluder dimension which is a complexity measure extensively used for bandits and reinforcement learning with function approximation. The paper studies the eluder dimension in relation to a new complexity measure proposed by the authors called $\sigma$-rank which is the minimal dimension required to represent the class as a generalized linear model.  The authors show that in some cases, the eluder dimension can be exponentially smaller than $\sigma$-rank. In some other cases where $\sigma$ is ReLu, the eluder dimension can be exponentially larger than $\sigma$-rank. Further, for binary-valued function classes, the authors provide a characterization of the eluder dimension in terms of star number and threshold dimension.


**Questions:**

I have some questions for the authors:
- Does the comparison between the $\sigma$-rank and the eluder dimension help understanding when the eluder dimension is bounded?
- In some cases, the eluder dimension can be exponentially smaller than $\sigma$-rank and otherwise. The authors can explain what is the consequence of this fact?
- When the function class lies in an RKHS, is there any equivalence between the eluder dimension and the generalized rank?

**Limitations:**

Yes

**Strengths And Weaknesses:**

Strengths:
- This paper addresses an important problem: understanding the insight of the eluder dimension in bandits and reinforcement learning with function approximation.
- The generalized notion of rank seems novel.  The results of the comparison of this notion and the eluder dimensions are interesting.
- The obtained results of this paper are original.

Weaknesses:
- The organization and the writing are not very clear to follow.
-  Although the paper addresses an insight into the eluder dimension, however, the questions raised in the Introduction section toward this insight are not convinced. Question 1 seems strange to me. Whether all function classes with small eluder dimensions are essentially generalized linear models is an "important" question? In [37], in a generalized linear setting, Russo and Van Roy provided an upper bound on the eluder dimension. This upper bound may be small or even infinite depending on the $\sigma$ function. Thus, we cannot require that the eluder dimension is small in generalized linear models. The answer "No" of Question 1 does not contrast with the result of Russo and Van Roy. I think that a more important question to answer is when the eluder dimension is bounded?
- Obtained results are not significant enough for the understanding of the eluder dimensions in existing works.

---

> ### Author Response · Authors · 2022-08-01
> **Thank you for your review.**
>
> We thank the reviewer for their comments and time. We make several important clarifications.
>
> **Regarding our Question 1.** We agree that the most important question to address is “when is eluder dimension bounded?” - this is exactly what this work seeks to answer. The reason why Question 1 is framed in terms of understanding the relationship between eluder dimension and generalized linear models is because this is a more manageable question to ask - the same way in which “when is VC dimension bounded?” is too broad compared to the question “what is the VC dimension of neural networks?”.
>
> The function classes whose eluder dimension we could compute an *upper bound* on were linear models and generalized linear models (with strictly monotone link function). Even though many papers have been published on the eluder dimension in the past decade, none have shown examples beyond (generalized) linear models for which one could compute an *upper bound* on eluder dimension.
>
> So, it is natural to wonder if this is truly an impossible task, to find an example of a function class which is not generalized linear for which we can upper bound the eluder dimension. Mathematically speaking, we ask if one can prove a *lower bound* on eluder dimension in terms of generalized linear rank. If this was true, then in conjunction with the upper bound of Russo and Van Roy, this would unequivocally show that *every function has bounded eluder dimension if and only if it is a generalized linear model (perhaps in disguise)*. This is the strongest possible answer one can get for the question of “when is eluder dimension bounded?”. It immediately implies that all the papers published in the past decade which bound regret in terms of eluder dimension can be rewritten as showing bounded regret assuming the function class is generalized linear.
>
> We show that this hypothetical “strongest possible” answer is indeed a pipe dream. Namely, Theorem 6 proves that there exists a simple function class (parities) where the eluder dimension is small but the generalized linear rank is quite large. Theorem 10 provides a stronger separation by constructing function classes for which the eluder dimension (with respect to a single base function) is constant but generalized linear rank is infinite. Therefore, it is impossible to prove a lower bound on eluder dimension in terms of generalized linear rank.
>
> Another restatement is that the set of function classes with bounded eluder dimension is strictly larger than the set of generalized linear function classes. Russo and Van Roy only show that the set of function classes with bounded eluder dimension contains the set of generalized linear function classes.
>
> *To conclude:* the answer of “no” to our Question 1, beyond the technical statement that “such a lower bound cannot hold”, provides a positive result for RL theory - it says that all these papers on eluder dimension actually apply to a *richer* set of functions classes than generalized linear models!
>
> **Regarding the connection to Russo and Van Roy when $\sigma$ is not strictly monotone.** You are correct in saying that the Russo and Van Roy upper bound becomes vacuous for generalized linear functions if $\sigma$ is not strictly monotone. Thus, for the relu function class the *upper bound* is infinite. Our contribution is to prove a *lower bound*: in Theorem 7, we show for the relu function class that the eluder dimension is at least exponential in the dimension. More generally, this shows that one cannot remove the dependence on $\mu$ in the bound for the eluder dimension of generalized linear models.
>
> We agree that this is not spelled out in the clearest way in the paper; our revision intends to make this discussion clearer.

---

> > ### Author Response · Authors · 2022-08-01
> > **Addendum.**
> >
> > Answering the reviewers questions:
> > 1. **Does comparing $\sigma$-rank and eluder dimension help us understand when eluder dimension is bounded? What is the consequence of eluder being exponentially smaller than $\sigma$-rank?** Yes, understanding the connection between $\sigma$-rank and eluder dimension allows us to show the existence of function classes (e.g., parities) which “go beyond” generalized linear. The main consequence is that there are function classes which have eluder dimension $d$, but the best possible bound that Russo and Van Roy’s result can give is $\exp(O(d))$.
> > The reason we carefully define the notion of $\sigma$-rank is that it allows us to be rigorous about what we mean when we say “a function class is generalized linear”. A priori, it is not obvious that parities cannot be rewritten as a generalized linear model in $\mathrm{poly}(d)$ dimensions. By defining the notion of $\sigma$-rank, we can formally say that the best possible bound that Russo and Van Roy can provide is $\exp(O(d))$.
> > A larger, less immediate consequence is that the results in bandits/RL literature which prove bounds via eluder dimension apply to a richer set of function classes than generalized linear models.
> > 2. **Function classes that lie in RKHS.** Function classes that lie in an (infinite dimensional) RKHS do not have finite dimension, so it is not meaningful to discuss their generalized rank. The recent note [1] proves that for function classes which do lie in an RKHS, the eluder dimension is equivalent (up to log factors) to the notion of information gain. This can be viewed as an extension of the bounds for eluder dimension for linear function classes (we know here that eluder dimension and information gain are exactly $\Theta(d \log R/\epsilon)$.
> >
> > [1] Huang, Kakade, Lee, Lei. “A Short Note on the Relationship of Information Gain and Eluder Dimension”.

---

### Official Review · Reviewer_82U6 · 2022-07-13

**Rating:** 7
**Confidence:** 3
**Soundness:** 3 good
**Presentation:** 3 good
**Contribution:** 3 good

**Summary:**

The authors study eluder dimension, its relationship with other measures like sigma-rk, and their binary versions. Specifically, they compare eluder/star dimension with various sigma-rk's and report their findings on which one is larger in general and when one can be much bigger than the other.

**Questions:**

I did not follow Eq (3). the condition implies that $\mu/L \le \frac{ \sigma(z')/L - \sigma(z)/L}{z'-z} \le 1$. The authors' claim makes sense if $\sigma$ is identity. Otherwise, I don't see how it is true; the range R is applied to the input of $\sigma$, but $L$ is on the gradient of $z$, so the the constant $R$ and $L$ are bounds for two different spaces.

other comments

* THeorem 7: should R here be $R^2$?
* What is Ldim? (littlestone dimension? I was not able to find the definition.)
* L308-309: why suddenly poly() here? The sentence here seems out of context..

**Limitations:**

None.

**Strengths And Weaknesses:**

(Please provide a thorough assessment of the strengths and weaknesses of the paper, touching on each of the following dimensions: originality, quality, clarity and significance. You can incorporate Markdown and Latex into your review. See /faq.)

Originality: moderate.

Quality: above bar

Clarity: above bar

Significance: moderate

The strength is the improved understand of various complexity measures. The weakness is that it is quite a theoretical work that does not have much implications about algorithms and the fundamental difficulty of learning problems.

-----
after the rebuttal period, I feel I have a better understanding of the contribution. Indeed, showing that EDim can be a strictly better complexity measure than sigma rank is quit meaningful and what researchers have been missing. Thus, I have raised the score.

---

> ### Author Response · Authors · 2022-08-01
> **Thank you for your review.**
>
> We thank the reviewer for their comments and time. Answering the questions in the review:
>
> 1. **Question on Eq (3)**. Here is a proof of Eq (3).
> Let $\sigma:\mathbb{R}\to \mathbb{R}$ be such that $\sigma \mathsf{-rk} (\mathcal{F}, R) = \mathcal{M}_\mu^L (\mathcal{F}, R)$. Then we define $\tilde{\sigma}(z) = \sigma(z/L)$.
> We can write every pair $(x,f)$ as $f(x) = \sigma(\langle w(f), \phi(x) \rangle) = \tilde{\sigma}(L \cdot \langle w(f), \phi(x) \rangle)$.
> We can compute that this $\tilde{\sigma} $ satisfies $\frac{ \tilde{\sigma} (z’) - \tilde{\sigma}(z) }{z'-z} = \frac{ \sigma(z’/L) - \sigma(z/L) }{z’- z} \in \[\frac{\mu}{L}, 1 \]$.
> This shows one direction; the other direction can be shown with a similar argument.
> 2. **On Theorem 7.** Yes, this is a typo. Thank you for catching it!
> 3. **What is Ldim?** Yes, Ldim is the Littlestone dimension. We apologize for omitting the definition; we will include it in later revisions. A formal definition can be found in the book [1], Definition 21.5.
> 4. **The poly() in L308-309.** We wanted to use this sentence to provide motivation for the next result. Theorem 8 proves an upper bound of $\mathsf{Edim} \le \mathrm{exp} (\max(\mathsf{Sdim}, \mathsf{Tdim}))$. However, this is just an inequality, and may not be tight. A priori, it could be possible using stronger techniques to prove an upper bound of $\mathsf{Edim} \le \mathrm{poly}(\mathsf{Sdim}, \mathsf{Tdim})$, for example $\mathsf{Edim} \le \mathsf{Sdim} \cdot \mathsf{Tdim}$ for any function class. However, Theorem 9 shows that this cannot be done, by exhibiting a specific function class where $\mathsf{Edim} \ge \exp ( \max (\mathsf{Sdim}, \mathsf{Tdim})).$
>
> We will gladly include these points in our revision.
>
> [1]: Shalev-Shwartz and Ben-David. “Understanding Machine Learning: From Theory to Algorithms.”

---

> > ### Comment · Reviewer_82U6 · 2022-08-08
> > **follow up**
> >
> > Makes sense! Thanks for your explanation.

---

### Official Review · Reviewer_4vAv · 2022-07-16

**Rating:** 7
**Confidence:** 3
**Soundness:** 4 excellent
**Presentation:** 4 excellent
**Contribution:** 3 good

**Summary:**


This paper gives an in-depth investigation on the notion of eluder dimension. Eluder dimension is a widely accepted complexity measure of function classes in bandits and reinforcement learning. However, it is previously unknown whether there is a separation between (function classes with) finite eluder dimension and generalized linear models. This paper is the first to show the separation.

 To show this, they defined $\Sigma$-rank, a new complexity measure that characterizes the generalized linear models. The paper first shows a chain inequality to link $\Sigma$-rank classes for different $\Sigma$. The paper then shows the place of Edim in the $\Sigma$-rank chain. Next, the paper narrows down the focus from real-valued functions to binary functions. Then the paper introduces combinatorial Edim, Sdim, Tdim, and shows an important tight characterization of Edim via Sdim and Tdim, via a connection to the Ramsey theory. At last, the paper asked if the results for binary (combinatorial) Edim can be brought back to real-valued functions, and they leave it as future work.


**Questions:**

N/A

**Strengths And Weaknesses:**


Strengths:

1. This paper identifies an important problem: is Edim different from generalized lienar models? Indeed, this is an important question that are less studied in the literature. Most papers just take the notion of Edim as granted and prove results based on Edim. But when asked about examples of Edim, they are elusive or just give generalized linear models as example.
2. The writing is good in this paper.

Weakness:

1. There are too many definitions of complexity measures in this paper. For better presentation, I would prefer them to be aggregated in a single section/appendix, which could help readers find and compare the definitions. It makes me hard to locate and figure out the definitions of different measures.
2. The paper might have less impact because recently, RL theory community are turning their focus from Edim to new definitions like Bellman rank and bilinear classes. Once people are proving results based on these two new defintions, it might be less interested to show results on Edim.
3. The proof are simple and combinatorial, which means it might not bring many new techniques or intuitions for future work.

---

> ### Author Response · Authors · 2022-08-01
> **Thank you for your review.**
>
> We thank the reviewer for their comments and time and have no corrections or objections.
>
> **A small comment regarding new definitions for function approximation**: Indeed, we are quite excited about these other notions for function approximation, like Bellman rank and bilinear classes. They allow one to prove interesting and general results. One distinction between the eluder dimension and these other measures of complexity is that eluder dimension is purely a property of the function class (and can be studied as such), while the other measures depend heavily on the underlying MDP (since they are defined in terms of the expected Bellman error under some roll-in policy in the given MDP).

---

### Meta-Review · Area_Chair_wFBw · 2022-08-23

**Recommendation:** Accept
**Confidence:** Certain

**Metareview:**

All reviewers and AC believe this paper is valuable contribution to the theoretical understanding of reinforcement learning.

**Award:**

No

---

### Decision · Program_Chairs · 2022-09-14

Accept